# Impact of CYP3A5 Polymorphisms on Pediatric Asthma Outcomes

**DOI:** 10.3390/ijms25126548

**Published:** 2024-06-14

**Authors:** Flory L. Nkoy, Bryan L. Stone, Cassandra E. Deering-Rice, Angela Zhu, John G. Lamb, Joseph E. Rower, Christopher A. Reilly

**Affiliations:** 1Department of Pediatrics, University of Utah School of Medicine, 100 N. Mario Capecchi Drive, Salt Lake City, UT 84113, USA; flory.nkoy@hsc.utah.edu (F.L.N.); bryan.stone@hsc.utah.edu (B.L.S.); angela.zhu@hsc.utah.edu (A.Z.); 2Department of Pharmacology and Toxicology, Center for Human Toxicology, University of Utah, 30 S 2000 E, Room 201 Skaggs Hall, Salt Lake City, UT 84112, USA; cassandra.rice@utah.edu (C.E.D.-R.); greg.lamb@pharm.utah.edu (J.G.L.); joseph.rower@hsc.utah.edu (J.E.R.)

**Keywords:** asthma, asthma control, asthma severity, glucocorticoids, inhaled corticosteroid, cortisol, Cytochrome P450 3A4, Cytochrome P450 3A5, pharmacokinetics, polymorphism

## Abstract

Genetic variation among inhaled corticosteroid (ICS)-metabolizing enzymes may affect asthma control, but evidence is limited. This study tested the hypothesis that single-nucleotide polymorphisms (SNPs) in Cytochrome P450 3A5 (CYP3A5) would affect asthma outcomes. Patients aged 2–18 years with persistent asthma were recruited to use the electronic AsthmaTracker (e-AT), a self-monitoring tool that records weekly asthma control, medication use, and asthma outcomes. A subset of patients provided saliva samples for SNP analysis and participated in a pharmacokinetic study. Multivariable regression analysis adjusted for age, sex, race, and ethnicity was used to evaluate the impact of CYP3A5 SNPs on asthma outcomes, including asthma control (measured using the asthma symptom tracker, a modified version of the asthma control test or ACT), exacerbations, and hospital admissions. Plasma corticosteroid and cortisol concentrations post-ICS dosing were also assayed using liquid chromatography–tandem mass spectrometry. Of the 751 patients using the e-AT, 166 (22.1%) provided saliva samples and 16 completed the PK study. The e-AT cohort was 65.1% male, and 89.6% White, 6.0% Native Hawaiian, 1.2% Black, 1.2% Native American, 1.8% of unknown race, and 15.7% Hispanic/Latino; the median age was 8.35 (IQR: 5.51–11.3) years. *CYP3A5*3/*3* frequency was 75.8% in White subjects, 50% in Native Hawaiians and 76.9% in Hispanic/Latino subjects. Compared with *CYP3A5*3/*3*, the *CYP3A5*1/*x* genotype was associated with reduced weekly asthma control (OR: 0.98; 95% CI: 0.97–0.98; *p* < 0.001), increased exacerbations (OR: 6.43; 95% CI: 4.56–9.07; *p* < 0.001), and increased asthma hospitalizations (OR: 1.66; 95% CI: 1.43–1.93; *p* < 0.001); analysis of *3/*3*, **1/*1* and **1/*3* separately showed an allelic copy effect. Finally, PK analysis post-ICS dosing suggested muted changes in cortisol concentrations for patients with the *CYP3A5*3/*3* genotype, as opposed to an effect on ICS PK. Detection of *CYP3A5*3/3*, *CYPA35*1/*3*, and *CYP3A5*1/*1* could impact inhaled steroid treatment strategies for asthma in the future.

## 1. Introduction

Asthma is the most common pediatric chronic illness in the United States (US), affecting ~4.7 million children <18 years of age [1]. In 2021, pediatric asthma accounted for 1.8 million asthma exacerbations, >270,000 emergency department (ED) visits and >27,000 hospitalizations [1]. In 2013, 13.8 million missed school days and 14.2 million missed workdays were attributed to asthma, leading to ~USD 3 billion in lost productivity [2]. The direct costs for pediatric asthma to the US healthcare system amount to ~USD 5.92 billion annually [3], and it is estimated that 60% of children with asthma have persistent asthma and could benefit from an ICS [4]. Despite the available evidence-based treatments for asthma, asthma control remains suboptimal in up to 63% of children [5,6,7,8,9].

ICSs are a mainstay treatment for controlling persistent asthma and preventing exacerbations [10]. Regular and appropriate use of an ICS reduces lung inflammation, bronchial hyper-responsiveness, exacerbations, ED/hospital visits, and overall asthma control [11,12,13]. However, asthma control varies among individuals despite proper use of their prescribed medications [14,15]. Multiple factors are associated with poor asthma control, [16,17,18,19,20,21,22,23,24,25,26,27,28,29] and polymorphisms among ICS-metabolizing enzymes [30,31] may contribute to the observation that ~50% of asthmatics have some degree of resistance (suboptimal response) or insensitivity (lack of response) to certain ICSs [32,33,34,35,36,37,38].

Genetic variations in Cytochrome P450 (CYP) enzymes are of particular interest because CYPs metabolize >80% of all prescribed drugs [39]. CYP3A enzymes (CYP3A4, 3A5, and 3A7) metabolize ~50% of medications [40], including ICSs. Prior work described roles for CYP3A4, 3A5, and 3A7 in ICS metabolism. CYP3A4 >> 3A5 > 3A7 were found to metabolize fluticasone propionate (FP), budesonide, flunisolide, and triamcinolone acetonide to the corresponding C6-oxygenated and C6-7-dehydrogenated metabolites, as well as 17- and 21-deesterified and/or carboxy metabolites, depending upon the substrate and enzyme [38]. FP also inhibited CYP3A5 [41]. Finally, beclomethasone dipropionate (BDP) was metabolized by CYP3A4 and 3A5 equivalently, but not by 3A7. The primary CYP metabolites of BDP were C6-hydroxy- and C6-7-dehydrogenated metabolites, while esterases catalyzed propionate ester hydrolysis into the pharmacologically active metabolite beclomethasone 17-monopropionate (BMP), and the inactive beclomethasone 21-monopropionate metabolite [42]. Cell-based metabolism assays showed that the CYP metabolites of BDP were only detected when esterases were inhibited and that ICSs universally induced the expression of *CYP3A5* in lung cells [37,43].

Within the *CYP3A5* gene, the **3* allele occurs most frequently (>90% in the White population), and codes for a non-functional enzyme, resulting in a poor metabolizer phenotype (as do the **6*, **7*, and other allelic variants). Within the *CYP3A4* gene, the *CYP3A4*22* variant codes for a non-functional protein. This variant occurs in <10% in the White population. We previously reported improved asthma symptom control among children with the *CYP3A5*3/*3* genotype for participants treated with BDP, and similar findings for children with the *CYP3A4*22* allele being treated with FP [44,45]. These initial studies were limited by the sample size, use of a less sensitive chronic asthma control measure that estimated average control over a one-year period [26], and researchers not controlling for several potential confounding factors. Also, the evaluation of *CYP3A5* genotype did not include other important asthma-related acute outcomes including exacerbations and ED/hospital admissions.

Here, the electronic AsthmaTracker (e-AT), a novel self-monitoring and management tool for children with asthma, was used to collect weekly assessments of asthma control status, asthma medication use and adherence, and acute asthma outcomes among participants enrolled at multiple pediatric ambulatory care clinics [46,47,48,49]. The e-AT provides a more accurate assessment of chronic asthma control over time, and to date, 751 children with asthma have used the e-AT as part of their standard of care. The objective of this study was to determine the impact of *CYP3A5* polymorphisms on asthma control and acute asthma-related outcomes in children using BDP, and to test the hypothesis that associations between asthma outcomes and *CYP3A5* polymorphisms may result from altered metabolism of BDP.

## 2. Results

### 2.1. Study Population

Of the 751 patients who used the e-AT between August 2012 and October 2022, 166 (22.1%) provided samples for genetic testing, with 16 completing the PK study. Enrollment of the participants for the PK study began in 2019, ended in 2022, and was severely limited due to COVID-19 restrictions. The demographic characteristics of the e-AT cohort were as follows: median age (8.35 yr; IQR: 5.51–11.30), male (108; 65.1%), female (58; 34.9%), White (149; 89.6%), Native Hawaiian (10; 6.0%), Black (2; 1.2%), Native American (2; 1.2%), unknown race (3; 1.8%), and Hispanic/Latino ethnicity (26; 15.7%). The distribution of the CYP3A5 genotype was as follows: 122 (73.5%) CYP3A5*3/*3, 39 (23.5%) CYP3A5*1/*3, and 5 CYP3A5*1/*1 (3.0%). All other CYP3A5 SNPs evaluated were present in <2% in our study population and were not pursued further. Demographic characteristics are presented in Table 1.

The 166 e-AT participants provided 7360 individual assessments of weekly asthma control status, for an average of 44.3 (SD = 55.2) weekly assessments per patient. Of the 7360 assessments of weekly asthma control, participants with the **3/*3* genotype completed 5128 assessments, those with the **1/*3* genotype completed 1995 assessments, and those with the **1/*1* genotype completed 237. The total number of OCS use entries was 814, including 482 in the **3/*3* group, 273 in the **1/*3* group, and 59 in the **1/*1* group. The total number of ED/hospital admission entries was 169, including 48 for the **3/*3* group, 100 for the **1/*3* group, and 21 for the **1/*1* group.

Due to the limited number of participants with the *CYP3A5*1/*1* genotype (n = 5), individuals with this genotype were combined with those with the *CYP3A5*1/*3* genotype for statistical analyses (i.e., the *CYP3A5*1/*x* genotype; n = 44). Therefore, in the primary analysis, we considered the outcomes of 122 participants with the *CYP3A5*3/*3* genotype alongside the outcomes of 44 participants with the *CYP3A5*1/x* genotype. For secondary analyses, the median of asthma control scores (with standard deviation) and the odds of having (or not having) asthma exacerbations or ED/hospital admissions during the one-year follow-up, were evaluated using data from 122 participants with the CYP3A5*3/*3 genotype 39 participants with the *CYP3A5*1/*3* and 5 participants with the CYP3A5*1/*1 genotype. All in all, 99 (59.6%) patients in total were prescribed FP alone or a FP/salmeterol combination, and 53 (31.9%) were prescribed BDP.

### 2.2. Asthma Outcomes and Genotype

Primary analysis using univariate statistical approaches identified a significant association for reduced asthma control (OR: 0.97; 95% CI: 0.97–0.98; *p* < 0.001), increased asthma exacerbations (OR: 6.07; 95% CI: 4.33–8.51; *p* < 0.001), and increased asthma-related hospitalizations (OR: 1.68; 95% CI: 1.45–1.96; *p* < 0.001) among participants with the *CYP3A5*1/*x* genotype compared with those with the *CYP3A5*3/*3* genotype. When controlled for covariates (age, sex, race, and ethnicity) and compared with *CYP3A5*3/*3*, the *CYP3A5*1/*x* genotype was significantly associated (Table 2) with reduced asthma control (OR: 0.98; 95% CI: 0.97–0.98; *p* < 0.001), increased asthma exacerbations (OR: 6.43; 95% CI: 4.56–9.07; *p* < 0.001), and increased asthma-related hospitalizations (OR: 1.66; 95% CI: 1.43–1.93; *p* < 0.001). Covariates were significant in multivariate analyses.

For secondary analysis (Table 3), the median asthma control score trended down from 20 (IRQ: 16–23) for *CYP3A5*3/*3*, 19 (IQR: 15–23) for **1/*3*, and 19 (IRQ: 16–22) for **1/*1*. Also, the odds of having (vs. not having) an asthma exacerbation were 0.009 (95% CI: 0.007–0.013) for *CYP3A5*3/*3*, 0.053 (95% CI: 0.043–0.065) for *CYP3A5*1/*3*, and 0.097 (95% CI: 0.062–0.152) for *CYP3A5*1/*1*. Similarly, the odds of experiencing (vs. not experiencing) asthma-related hospitalization were 0.104 (95% CI: 0.094–0.114) for *CYP3A5*3/*3*, 0.158 (95% CI: 0.139–0.180) for *CYP3A5*1/*3*, and 0.331 (95% CI: 0.247–0.445) for *CYP3A5*1/*1*.

### 2.3. ICS PK Analysis

Blood was collected from a total of 16 participants for PK analysis. Eight used BDP and eight used FP; nine were male and seven were female. The median age was 11 yr, with a range of 5–18 yr. In total, 6 participants had the *CYP3A5*1/*x* genotype and 10 had the *CYP3A5*3/*3* genotype. Among the eight participants in the BDP-treated group, only three (two male and one female; median age: 11 yr) had the *CYP3A5*1/*x* genotype; the remaining 5 BDP-treated patients had the *CYP3A5*3/*3* genotype (three male and one female; median age: 9.5 yr). Notably, one of the BDP-treated participants with the *CYP3A5*3/*3* genotype (18 yr old male) also had the *CYP3A4*1/*22* genotype. Only 3 of the 48 total plasma samples had quantifiable concentrations of BDP. Thus, PK analyses of BDP were not possible. However, concentrations of BMP normalized to a 40 mg dose of BDP are shown in Figure 1. Though appropriately powered comparisons could not be performed, the data revealed interesting and potentially meaningful insights. Initial peak concentrations of BMP were achieved rapidly and were comparable for patients with the **1/*x* and **3/*3* genotypes, at 304 and 243 pg/mL, respectively. The mean dose-normalized area under the concentration–time curves (AUC) were identical (1.44 [**1/*x*] vs. 1.44 [**3/*3*] pg·h/mL/40 mg). A calculation of clearance was not performed due to the presence of a secondary peak that did not begin to be eliminated until after the final blood sample was collected at 8 h. This secondary peak was particularly evident among individuals with the *CYP3A5*3/*3* genotype. While the cause of this secondary peak remains unknown, we hypothesize that it may have resulted from a secondary route of absorption (oral) following incomplete delivery of the aerosolized spray into the airways, which is a common occurrence when using ICSs. FP concentrations were also undetectable in most plasma samples, preventing PK analysis.

### 2.4. Cortisol PK Analysis

As expected, cortisol concentrations decreased following the administration of BDP (Figure 2). Baseline mean cortisol concentrations were 72.5 and 93.5 ng/mL for participants with the **1/*x* and **3/*3* genotypes, respectively. At the 4 h time point, the mean cortisol concentrations were 18.6 and 23.4 ng/mL for the **1/*x* and **3/*3* genotypes, respectively, with cortisol concentrations for the *1/*x genotype remaining lower (suppressed) up to 8 h compared with those of the participants with the *CYP3A5*3/*3* genotype. The most pronounced difference was observed between 4 and 6 h. At 6 h, the mean cortisol concentrations were 32.1 versus 58.8 ng/mL for the **1/*x* and **3/*3* genotypes, respectively. Though the sample size was insufficient for statistical analysis, a comparison of cortisol recovery (i.e., the proportional increase in cortisol concentration from 4 to 6 h relative to time 0 of each individual) revealed an 11.11 ± 0.06 versus 39.1 ± 0.2% increase for those with the *CYP3A5*1/*x* and **3/*3* genotypes, respectively. Of note, the patient with the *CYP3A5*3/*3* + *CYP3A4*1/*22* genotype (not included in the analysis above) showed the greatest cortisol recovery: 21.2 to 69.4 ng/mL from 4–6 h compared with 43.9 at time = 0. When included in the *CYP3A5*3/*3* group, a mean recovery of 53.2 ± 0.3% was observed. The study protocol for FP was limited to 5 h; thus, cortisol recovery among FP users could not be evaluated.

## 3. Discussion

The results demonstrate that the *CYP3A5*3/*3* genotype is associated with improved asthma outcomes, including improved symptom control, fewer asthma exacerbations, and fewer asthma-related ED/hospital admissions. When compared with the reference *CYP3A5*3/*3* group, individuals with ≥1 **1* allele had a 2% decrease in mean asthma control scores, 6-fold greater odds of OCS use (representing asthma exacerbations), and 1.7-fold greater odds of asthma-related ED/hospital admissions. These associations were maintained in analyses that controlled for demographic factors including age, sex, race, and ethnicity, as well as the prescribed ICS. When the odds of having asthma outcomes were evaluated, we found consistent trends, suggesting an allele copy effect. Individuals with the *CYP3A5*3/*3* genotype had the most favorable outcomes, followed by those with *CYP3A5*1/*3*, then *CYP3A5*1/*1*. Finally, PK analyses suggested that improved outcomes could result from genotype-dependent differences in cortisol PK post-BDP dosing, providing potential rationale for the current and our previously reported [44,45] clinical findings on CYP3A polymorphisms and asthma control.

Understanding the mechanisms linking *CYP3A* genotypes with clinical asthma severity could inform improved clinical care for asthma. We previously reported that BDP was preferentially metabolized and inactivated by CYP3A5, whereas CYP3A4 was dominant for FP and most other commonly used ICSs [38,42,45]. Like inhibition, polymorphisms can affect the quantity of functional enzyme and the metabolism of drugs, including BDP [50]. *CYP3A5*3* codes for an inactive enzyme [51]. Thus, individuals with the *CYP3A5*3/*3* genotype are considered poor metabolizers, with a non-functional enzyme that could limit the metabolic clearance of BDP and other ICSs in the gut, liver, and lung. Limiting ICS metabolism could enhance efficacy [40]. On the other hand, individuals with *CYP3A5*1/*3* and *CYP3A5*1/*1* genotypes are considered intermediate and “normal” metabolizers. Thus, *CYP3A5*1* could enhance ICS clearance and attenuate ICS efficacy [50], particularly with BDP. A similar rationale applies for *CYP3A4*1* and *CYP3A4*22*, the latter of which is a non-functional enzyme that affects CYP3A4 activity in the intestines and liver.

ICS therapy also reduces adrenal cortisol secretion [52,53,54], with high doses of FP, and to a lesser extent BDP, causing the greatest suppression in this drug class [55]. Here, participants with the *CYP3A5*3/*3* genotype had both improved asthma outcomes and evidence of muted cortisol suppression and a more rapid return to basal levels post-BDP dosing, with no obvious effect on BMP metabolism. This led us to hypothesize that cortisol maintenance post-BDP dosing may impact asthma control, as opposed to having a direct effect on BDP/BMP PK. Specifically, the *CYP3A5*3* (and *CYP3A4*22*) alleles alone or in combination may decrease cortisol 6β-hydroxylation (i.e., an inactive metabolite), regardless of CYP3A4/5 enzyme induction associated with corticosteroid use, thus decreasing the degree of suppression and allowing for a more rapid restoration of “normal” levels post-ICS dosing. Cortisol is the endogenous anti-inflammatory agent that corticosteroids mimic. Cortisol and ICSs control asthma/inflammation through the suppression of pro-inflammatory genes (e.g., NF-kB and AP-1 regulated genes) and the activation of the transcription of anti-inflammatory genes such as *IL10*, *IL12*, and *IL1RN* [56]. The net effect is generally reduced eosinophil, T-cell, mast, and dendritic cell numbers, particularly in the lungs. While the exact role of cortisol in asthma is not yet fully resolved [57,58], reduced cortisol may contribute to nocturnal airway obstruction [59] and poorer asthma control [60]. Accordingly, maintaining “normal” cortisol levels in the context of ICS therapy may be important for optimal asthma control, with *CYP3A4/5* genetics being one factor regulating the cortisol–corticosteroid balance.

Studies have also reported significant disparities in asthma outcomes, with asthma burden falling disproportionately on low-income patients and minorities [61]. Black, Hispanic/Latino, and Native American/Alaska Native patients have higher rates of exacerbations, ED visits, hospitalizations, and deaths than White patients [61,62]. Hospitalization and mortality rates for Black patients are six times and three times higher, respectively, than those for White patients [63]. Although differences in social determinants of health can partly explain these disparities [64,65], variations in the racial/ethnic distribution of *CYP3A5*1* and **3* may also underly these findings. Specifically, the frequency of the *CYP3A5*1* allele is higher in non-White populations, indicating that current ICS dosing regimens could be further optimized in these populations by developing and implementing personalized dosing regimens for individuals with *CYP3A5*1/*x* genotypes, and controlling for cortisol suppression/recovery.

This study has several limitations, and the results need to be interpreted with caution. First, the data used came from patients using the e-AT. Prior studies showed that patients with severe asthma were more likely to use the e-AT vs. patients with mild or moderate asthma. Thus, the results could change if more patients with moderate asthma were included. Second, participants were recruited from 35 different clinics in Utah. Although the e-AT guides physicians in identifying patients’ needs early, and in providing evidence-based asthma treatment, differences in asthma management among participating clinics could impact the results. Third, asthma control and outcomes are affected by multiple factors such as patient and provider failure to recognize and act on early signs of declining asthma control, lack of patient self-management skills, patient non-adherence with therapy, inappropriate ICS prescription by providers, differences in social determinants of health, differential exposures to asthma triggers including air pollution, etc. [16,17,18,19,20,21,22,23,24,25,26,27,28,29]. Although the current analyses were controlled for age, sex, and race/ethnicity, many other factors were not controlled and could have impacted the results. Fourth, the study participants lacked diversity and do not necessarily reflect the racial/ethnic diversity of the US asthma population. Although ~16% of participants identified as Hispanic/Latino, nearly matching the 15% of Utah’s representation for Hispanic/Latino people (2020 Utah Census), the results could differ if the study population was more diverse, and the generalizability of the results improved. Fifth, outcome data were captured only when patients used the e-AT. Thus, the results could be impacted by outcome reporting bias. We do not believe this is an issue since the current results generally agree with our prior studies utilizing different measures of asthma control [44,45], indicating that CYP3A genotypes, and *CYP3A5*3/*3* in particular, are indeed associated with improved asthma control. Sixth, the current study found that the *CYP3A5*3/*3* genotype was associated with improved asthma control regardless of FP or BDP use. These results contradict our prior studies where we found that the effects of *CYP3A5*3* occurred only among BDP-treated patients [44], and those of *CYP3A4*22* involved FP treatment. This difference may be due to the inconsistent recording of ICS use in the eAT by patients, or that the prior study did not accurately document medication changes. Regardless, the current findings suggest a broader, generic role for CYP3A enzymes in the regulation of cortisol–corticosteroid dynamics, as discussed above. Sixth, we did not assess at baseline, patients with chronic OCS treatment, and the inclusion of such patients could affect our results. Finally, the PK study was limited in sample size, and we did not tightly control the initiation of the PK study, which could have impacted findings regarding genotype and cortisol PK. However, we do not believe this was a major issue since all participants were dosed in the morning when cortisol levels are naturally lower, and the analysis normalized data to the pre-dose time = 0 cortisol value, presumably minimizing the impact of individual differences and the daily rhythm of cortisol. Accordingly, a future study with more accurate recording of asthma medication uses and ICS dosing is needed to determine whether or not the effects of *CYP3A5* polymorphisms apply to other corticosteroids and to verify the trends identified here.

To summarize, the *CYP3A5*3/*3* genotype was associated with improved asthma outcomes, and the effect was not specific to the prescribed ICS. Rather, the effect appeared to be a general effect of CYP3A5 (and 3A4) deficiency, potentially involving the maintenance of a more “normal” cortisol level post-ICS use, as opposed to a direct effect on ICS PK. More studies are needed to fully understand the mechanisms underlying the effect of the *CYP3A5*3/*3* genotype and other CYP polymorphisms on asthma outcomes, but it is apparent that accounting for such variations could guide the selection and utilization of existing asthma control medications to improve asthma control on an individual basis.

## 4. Materials and Methods

### 4.1. Setting and Study Population

Study participants were enrolled from 35 ambulatory pediatric clinics that implemented the e-AT to remotely monitor and manage patients with asthma. The cohort consisted of children aged 2–18 years with a physician diagnosis of persistent asthma. Study procedures were approved by the University of Utah IRB. Parental permission and authorization (from legal guardians of all children) and assent (for children >7 years of age) were obtained for eligible participants after either an in-person or telephone discussion of the informed consent/assent documents, planned study procedures, and potential risks to the participant. Consent/assent documents were signed in person prior to initiating study procedures, with additional details below.

### 4.2. e-Asthma Tracker

The e-AT is a web and mobile web application designed to support home monitoring and management of asthma in children [47]. The application can be used by an adolescent or a primary caregiver of a younger child to monitor asthma control weekly. Users receive immediate feedback and recommendations when early signs of deterioration of asthma control are detected, prompting proactive interventions and/or more timely care by the primary care providers (PCPs), thereby preventing ED/hospital admissions [46,49]. In addition to clinical data, the e-AT database includes participant demographic information including age, sex, race (categorized as White, Black, Asian, Native Hawaiian or Other Pacific Islander, Native American, and Unknown), and ethnicity (Hispanic/Latino, non-Hispanic/Latino and Unknown).

The e-AT includes a patient interface and a web-based clinic dashboard. The patient interface prompts a weekly self-assessment of asthma control using the Asthma Symptom Tracker (AST), a modified version of the Asthma Control Test (ACT) [66] validated for the weekly assessment of asthma control by patients or caregivers for younger patients [48]. Similar to the standard ACT [66], asthma control scores on the AST range from 5 (poor control) to 25 (optimal control), allowing categorization into three groups: 19–25 (well-controlled), 14–18 (not well-controlled) and <14 (poorly controlled). The e-AT also collects data on asthma medication use/adherence (although it does not guarantee that a patient used their medications properly or as prescribed) and asthma-related acute outcomes, including whether or not the child had an asthma exacerbation requiring oral corticosteroids (OCS) or an ED/hospital visit during a specific week. Other features of the e-AT include automated reminders to support adherence with weekly use; real-time patient feedback with longitudinal graphs, categorization of asthma control as well-, not well-, or poorly controlled; real-time alerts to patients or caregivers (via email or text) and PCPs (via email or clinic dashboard); real-time recommendations based on asthma control category; and motivational features to encourage regular use. The clinic dashboard allows efficient asthma population management, including ready access to patient’s real-time asthma control statuses, longitudinal control graphs, medication use adherence, and alerts to guide treatment adjustments.

### 4.3. Study Procedures and DNA Collection and Analysis

Participants were trained on how to use the e-AT by the clinic care coordinators and were asked to use the e-AT weekly as part of their routine asthma care. They were contacted by a research coordinator by phone or email, and if they consented, were sent a saliva sample kit with instructions for sample collection and pre-paid return postage. Enrollees were also offered the opportunity to participate in a clinic visit where they would use their prescribed ICS with follow-up blood draws to assess plasma ICS and cortisol concentrations.

The collection of genomic DNA (gDNA) and genotyping analyses have been described previously [44]. Similar collection and genotyping methods were utilized for participants described in these analyses. Saliva or buccal swabs (younger children) were collected using established protocols with a sterile DNA/RNA Shield^TM^ collection kit (Zymo Research, Irvine, CA, USA) pre-filled with a DNA-stabilizing agent. All samples were labeled with a unique patient ID number and the date and time of collection and delivered to the laboratory for analysis. All patients received a monetary incentive for providing their saliva/buccal samples.

Genomic DNA was isolated from 2 mL of the samples using the PureLink gDNA mini kit (ThermoFisher, Carlsbad, CA, USA) per the manufacturer’s protocol. *CYP3A5* allele status was determined from 10 ng of gDNA using the TaqMan SNP genotyping enzyme (ThermoFisher, Carlsbad, CA, USA) and probes for rs776746 (assay ID: C_26201809_30) and several others (i.e., *CYP3A5*1D*/rs15524, **5*/rs55965422, **6*/rs10264272, **7*/rs41303343, **8*/rs55817950, **9*/rs28383479, and K208K/rs10264272). *CYP3A4*22* (rs35599367) allele status was also determined using assay ID C_59013445_10. TaqMan reactions were cycled as recommended by the manufacturer on a Life Technologies QuantStudio6 instrument. Data clustering analysis was performed using TaqMan Genotyper software (v1.3.1; ThermoFisher, Carlsbad, CA, USA).

### 4.4. PK Study Enrollment and Procedures

A separate consent process was completed for participants willing to have blood collected for PK analysis. On the morning of their scheduled PK study visit, participants were requested to bring their prescribed ICS to the University of Utah Clinical Translational Science Institute (CTSI), and to refrain from using the ICS until directed by CTSI nursing/phlebotomy staff. All participants had a single dose of their prescribed ICS dose administered in the morning, typically between 7:00 and 10:00 a.m., and approximately 12 h or more after their previous ICS dose. A pre-dose sample was collected for all participants using a peripheral venous catheter. The participant was then directed to administer their prescribed ICS dose, followed by blood sample collection at either 0.75, 2, 4, 6, and 8 h (BDP; Vacutainer Li-Heparin tubes) or 0.25, 1, and 5 h (FP; Vacutainer NaF/K-oxalate tubes) post-dose. Blood was immediately placed on ice and centrifuged at 4 °C, and the plasma was stored at −80 °C until analysis.

### 4.5. Bioanalytical Assay

BDP, BMP, FP, and cortisol were quantified using a validated liquid chromatography–tandem mass spectrometry (LC-MS/MS) method. Reference standards and their deuterated internal standards were purchased from Sigma Aldrich. Internal standards in methanol were added to a 250 μL serum aliquot, 0.5 mL of 10% (*v*:*v*) ammonium hydroxide was added to dilute the plasma, and the analytes were extracted into 2.0 mL of methyl *tert*-butyl ether via vortex-mixing, centrifugation, freezing, and decanting the organic layer into clean polypropylene tubes. The organic layer was then dried in room air in Zymark TurboVap (Hopkinton, MA, USA; 15 psi, 40 °C) and reconstituted in 50:50 methanol:water (50 mL).

LC-MS/MS analysis utilized ThermoScientifc TSQ Vantage (ThermoScientific, San Jose, CA) operated in positive electrospray ionization mode interfaced with an Accela autosampler and pump. Chromatographic separation was achieved with isocratic elution using 70:30 methanol:5 mM ammonium bicarbonate (pH 8) at 0.25 mL/min on a Phenomenex (Torrence, CA, USA) Luna Omega PS C_18_ 100 × 2.1 mm (3 μm particle size) column held at 40 °C. Mass transitions (collision energy, CE) for BDP, BMP, FP, and cortisol were 521.3 → 301.3 (20 V), 465.1 → 279.1 (20 V), 501.2 → 274.2 (30 V), and 363.1 → 121.0 (25 V), respectively. Calibration curves were fit with a 1/x^2^ weighted linear regression and ranged between 10–2500 pg/mL (BDP), 50–5000 pg/mL (BMP), 20–5000 pg/mL (FP), and 0.16–80 ng/mL (cortisol). Cortisol calibrators and quality control samples were prepared in 5% bovine serum albumin in a phosphate-buffered saline solution due to the presence of cortisol in plasma; plasma calibrators and quality control samples were used for all other analytes.

### 4.6. Statistical Analysis

Baseline patient characteristics were compared using the mean, standard deviation (SD), and Wilcoxon rank sum (Mann–Whitney) test for continuous variables, counts, and frequencies. The X-Square test was used for categorical and ordinal variables.

Univariate linear regression analysis was used to evaluate associations between *CYP3A5* genotypes and individual asthma outcomes, including asthma control, asthma exacerbations, and asthma-related ED/hospital admissions. Weekly asthma control scores were Log-transformed to reduce skewness and used as a continuous variable. OCS use (a surrogate measure of asthma exacerbation) and hospital admissions were dichotomized (Y/N) and used as the dependent variable in the analysis. Finally, multivariate linear and logistic analysis was performed to test for associations after controlling for participant age, sex, race, and ethnicity, and prescribed ICS (FP or BDP). To facilitate analysis, and because of the small sample size, *CYP3A5*1/*1* was combined with the *CYP3A5*1/*3* genotype to create the category *CYP3A5*1/*x*. Asthma outcomes as a function of *CYP3A5*3/*3* (used as a reference for all analyses) and *CYP3A5*1/*x* were compared. The impact of *CYP3A5*3/*3*, *CYP3A5*1/*3*, and *CYP3A5*1/*1* were also evaluated separately to assess whether there was an allelic copy effect in the association between the *CYP3A5* genotype and asthma outcomes. For the primary analysis (comparing **3/*3* with **1/*3* and with **1/*1*), we used a linear model (the Log-transformed asthma control score) to estimate β coefficients (i.e., the effect size), which were then converted into odds ratios (ORs) and the associated 95% confidence intervals (CIs). For logistic models (OCS use and asthma admission), ORs and 95% CIs, along with *p*-values, were determined. For secondary analysis, we reported the median asthma control scores with the interquartile range (IQR) and the odds of having (or not having) asthma exacerbations or hospital admissions separately for each genotype. All analyses were conducted using Stata/IC version 16.1.

## 5. Conclusions

The detection of *CYP3A5*3/3*, *CYPA35*1/*3*, and *CYP3A5*1/*1* could impact inhaled steroid treatment strategies for asthma in the future.

## Figures and Tables

**Figure 1 ijms-25-06548-f001:**
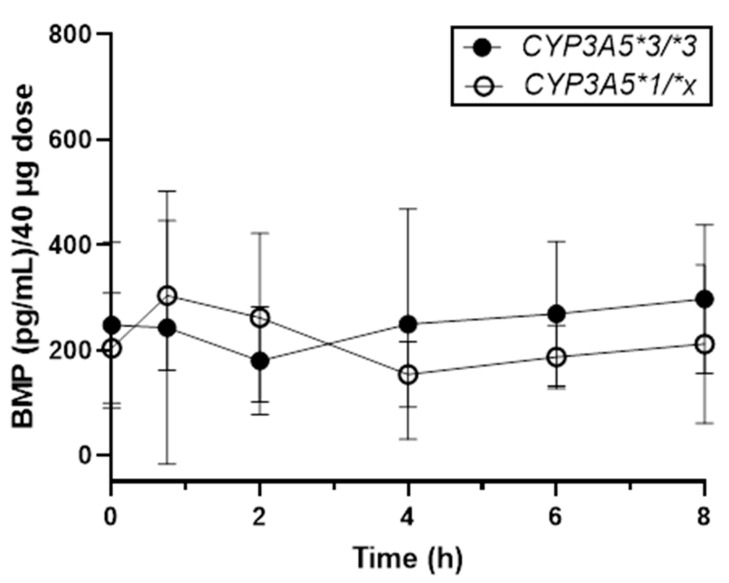
Concentration vs. time profile of plasma BMP for patients with the *CYP3A5*3/*3* vs. *CYP3A5*1/*x* and *CYP3A4*1/*1* genotypes. Data represent the mean ± standard deviation of *n* = 5 and *n* = 3 participants with the *CYP3A5*3/*3* (closed circles) and *CYP3A5*1/*x* (open circles) genotypes, respectively.

**Figure 2 ijms-25-06548-f002:**
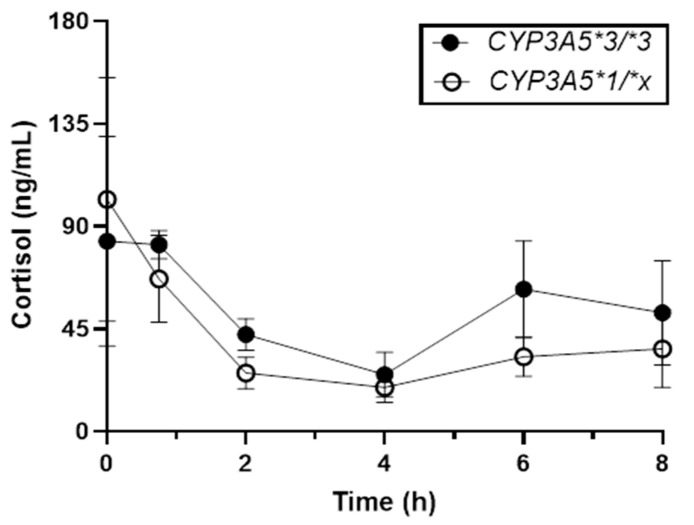
Concentration vs. time profile of plasma cortisol for patients with the *CYP3A5*3/*3* vs. *CYP3A5*1/*x* and *CYP3A4*1/*1* genotypes. Data represent the mean ± standard deviation of *n* = 5 and *n* = 3 participants with the *CYP3A5*3/*3* (closed circles) and *CYP3A5*1/*x* (open circles) genotypes, respectively.

**Table 1 ijms-25-06548-t001:** Study population characteristics.

Criteria	Category	Median (IQR) or *n*	*CYP3A5*3/*3*	*CYP3A5*1/*x*	*p*-Value
Age (yr)	n/a *	8.35 (5.51–11.3)	8.0 (5.0–11.0)	9.0 (7.0–11.5)	0.242
Sex	Male Female	108 58	77 45	31 13	0.381
Race	White American Indian Black Native Hawaiian Unknown	149 2 2 10 3	113 1 1 5 2	36 1 1 5 1	0.347
Ethnicity	Hispanic Latino Non-Hispanic/Latino Unknown	26 137 3	20 100 2	6 37 1	0.884
FP-treated	Yes No	99 67	73 49	26 18	0.931
BDP	Yes No	53 113	42 80	11 33	0.250
BDP-treated	Yes No	6 160	6 116	0 44	0.134

* n/a = not applicable.

**Table 2 ijms-25-06548-t002:** Comparison of asthma outcomes between *CYP3A5*3/*3* and *CYP3A5*1/x* genotypes.

Outcomes	Genotype	OR	95% CI	*p*-Value
Asthma Control	*CYP3A5*3/*3*	1	1	1
	*CYP3A5*1/*x*	0.98	0.97–0.98	<0.001
Asthma Exacerbations	*CYP3A5*3/*3*	1	1	1
	*CYP3A5*1/*x*	6.43	4.56–9.07	<0.001
Hospital Admissions	*CYP3A5*3/*3*	1	1	1
	*CYP3A5*1/*x*	1.66	1.43–1.93	<0.001

**Table 3 ijms-25-06548-t003:** Asthma outcomes for each CYP3A5 genotype.

Outcome	Genotype	Median	Interquartile Range
Asthma Control	*CYP3A5*3/*3*	20	16–23
	*CYP3A5*1/*3*	19	15–23
	*CYP3A5*1/*1*	19	16–22
**Outcome**	**Genotype**	**Odds**	**95% CI**
Asthma Exacerbations	*CYP3A5*3/*3*	0.009	0.007–0.013
	*CYP3A5*1/*3*	0.053	0.043–0.065
	*CYP3A5*1/*1*	0.097	0.062–0.152
Hospital Admissions	*CYP3A5*3/*3*	0.104	0.094–0.114
	*CYP3A5*1/*3*	0.158	0.139–0.180
	*CYP3A5*1/*1*	0.331	0.247–0.445

## Data Availability

The authors declare that all the data supporting the findings of this study are contained within the paper. Raw data may be made available upon request but may be subject to restrictions due to IRB and/or material transfer agreement requirements.

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
