# Peer review of "Impact of CYP3A5 Polymorphisms on Pediatric Asthma Outcomes"

_ijms, 2024, doi:10.3390/ijms25126548_

Round 1
Reviewer 1 Report
Comments and Suggestions for Authors
This is an interesting study on genetic variants that have impact on asthma control. The manuscript is well written, novel and important as it migght show new ways of planning inhaled asthma treatment in the future. As a practicing physician I look forward for more research in this area possibly leading to new treatment personalized guidelines.
The paper has some minor issues though that should be adressed. Please find the comments below.
Abstract
There should be information about primary outcomes in abstract, so 1)AST as a modified version of ACT should be mentioned, 2)hospital admisions, 3) exacerbations. It is important for physician what tools you use in asthma course assessment and it should be known from the beginning.
Methods
Line 402 why was not T-test used for independent continuous variables?
I am missing information on 1) How long did a single participant be in the study? 2) How was the informed consent obtained? The second one is very important since it is a pediatric population.
Since asthma exacerbation was described as use of OCS I understand that patients on chronic OCS treatment were not included? This should be clarified. Also, clear definition of asthma exacerbation according to the study should be provided in Methods and under table descriptions.
Discussion
Sentence line 288 ". Finally, the PK study was limited in power/sample size and did not tightly control the initiation of the PK study, which could have impacted our findings regarding genotype and cortisol PK." is difficult to understand and might have some grammar issues.
Line 293 Why is circadian capital? Suggest changing to more simple word like "daily rhytm"
Conclusions
Though interesting, still I believe conclusions go a little too far at this point. Consider changing conclusions to something like: "Detection of CYP3A5*3/*3, CYP3A5*1/*3, and CYP3A5*1/*1 could impact asthma inhaled steroid treatment strategies in the future."
Author Response
ABSTRACT
There should be information about primary outcomes in abstract, so 1) AST as a modified version of the ACT should be mentioned, 2) hospital admissions, and 3) exacerbations. It is important for physician what tools you use in asthma course assessment and it should be known from the beginning.
Response: We added information about primary outcomes to the abstract (lines 19-22) as requested.
METHODS
Line 402 why not T-test used for Independent continuous variables?
Response: Because of skewness in asthma control data, the T-test was not appropriate.
I am missing information on 1) How long did a single participant be in that study? 2) How was the informed consent obtained? The second one is very important since it is a pediatric population.
Response: Patients were already using the e-AT as part of quality improvement at 11 clinics that lasted 12 months. Since they initiated the e-AT as part of quality improvement, no consent was obtained. The consent was limited to those who agreed to provide a saliva sample for genetic testing for the current study (to compare genetic polymorphisms and their asthma outcomes). Participants were enrolled for 1y, but the study encompassed multiple years of data collection. For the optional PK study, participants were on study for up to 8h, with post-visit monitoring for adverse events for 48h.
Regarding consent, the following text has been added to the Methods section, 1st paragraph (lines 315-320): “Parental permission and authorization (from legal guardians of all children) and assent (for children >7 yr of age) were obtained for eligible participants after either in-person or telephone discussion of the informed consent/assent documents, planned study procedures, and potential risks to the participant. Consent/assent documents were signed in person prior to initiating study procedures, with additional details below.”
Since asthma exacerbation was described as use of OCS. I understand that patients on chronic OCS treatment were not included? This should be clarified. Also, clear definition of asthma exacerbation according to the study should be in Methods and under table descriptions.
Response: We did not collect this information and added this as a limitation in the discussion section, lines 291-292.
DISCUSSION
Sentence line 288 “. Finally the PK study was limited in power/sample size and did not tightly control the initiation of the PK study, which could have impacted our findings regarding genotype and cortisol PK.” is difficult to understand and might have some grammar issues.
Response: We made changes and hope the sentence is now clear.
Line 293 Why is circadian capital? Suggest changing to more simple word like “daily rhythm”
Response: This was an error. Regardless, we changed the wording to “daily rhythm of cortisol” as suggested.
CONCLUSIONS
Though interesting, still I believe conclusions go a little too far at this point. Consider changing conclusions to something like: “Detection of CYP3A5*3/3, CYPA35*1/*3, and CYP3A5*1/*1 could impact asthma inhaled steroid treatment strategies in the future.”
Response: We made changes to the conclusion of the abstract and section 5 of the manuscript as suggested.
Reviewer 2 Report
Comments and Suggestions for Authors
The research on the impact of genes polymorphism on asthma course and the response to treatment is a very interesting and constantly developing topic. In this context the paper of Nkoy and colleagues poses a very valuable input into this area of knowledge. The Authors combine 2 very modern approaches: molecular science and electronic monitoring with the ultimate goal of improving the outcomes in children and adolescents with asthma.
My only concern with this paper would be, that while the e-AT includes automated reminders to support adherence with medication use it does not actually check whether the patients took their medication. And since drug concentrations were measured in only few participants we have no means to evaluate how many patients took their drugs at all and how many in their prescribed doses. Non-compliance probably happened independently of the genotype as it happens in real life but this should be just considered and mentioned a minor limitation of the study.
Author Response
My only concern with this paper would be, that while the e-AT includes automated reminders to support adherence with medication use it does not actually check whether the patient took their medication. And since drug concentrations were measured in only few participants we have no means to evaluate how many patients took their drugs at all and how many in their prescribed.
Response: Correct, the e-AT only collects data for whether a patient used their medication and how many days they used it during a week. It does not actually measure whether the patient used their medication as directed/prescribed through plasma drug analysis. However, for the PK part of the study, the study coordinator and clinic staff observed patients using their medication.
We have modified the sentence starting on line 336 to read: “The e-AT also collects asthma medication use/adherence (although it does not guarantee that a patient used their medications properly or as prescribed) and asthma related acute outcomes including whether the child had an asthma exacerbation requiring oral corticosteroids (OCS) or an ED/hospital visit during a specific week.”